# *Bitis arietans* Snake Venom Induces an Inflammatory Response Which Is Partially Dependent on Lipid Mediators

**DOI:** 10.3390/toxins12090594

**Published:** 2020-09-14

**Authors:** Angela Alice Amadeu Megale, Fernanda Calheta Portaro, Wilmar Dias Da Silva

**Affiliations:** Immunochemistry Laboratory, Butantan Institute, São Paulo 05503-900, Brazil; angela.amadeu@butantan.gov.br

**Keywords:** *Bitis arietans* venom, inflammatory process, lipid mediators, genetically deficient mice

## Abstract

*Bitis arietans* is a snake of medical importance, as it is responsible for more accidents in humans and domestic animals than all other African snakes put together. The accidents are characterized by local and systemic alterations, such as inflammation, cardiovascular and hemostatic disturbances, which can lead victims to death or permanent disability. However, little is known about the envenomation mechanism, especially regarding the inflammatory response, which is related to severe clinical conditions triggered by the venom. Therefore, the aim of the present study was to evaluate the inflammatory response related to the *B. arietans* envenomation using a peritonitis mice model. By pharmacological interventions and use of mice genetically deficient of the 5-lipoxygenase enzyme (5-LO^−/−^) or platelet-activating factor (PAF) receptor (PAFR^−/−^ the participation of eicosanoids and PAF in this response was also investigated. The obtained results demonstrated that the venom induces an in vivo inflammatory response, characterized by an early increased vascular permeability, followed by an accumulation of polymorphonuclear (PMN) cells in the peritoneal cavity, accompanied by the production of the eicosanoids LTB_4_, LTC_4_, TXB_2_ and PGE_2_, as well as the local and systemic production of IL-6 and MCP-1. These inflammatory events were attenuated by the pre-treatment with anti-inflammatory drugs that interfere in lipid mediators’ functions. However, 5-LO^−/−^ mice did not show a reduction of inflammatory response induced by the venom, while PAFR^−/−^ mice showed a reduction in both the PMN leukocytes number and the local and systemic production of IL-6 and MCP-1. This study demonstrated that the *Bitis arietans* venom contains toxins that trigger an inflammatory process, which is partially dependent on lipid mediators, and may contribute to the envenomation pathology.

## 1. Introduction

Snakebite is a serious and neglected public health problem worldwide, especially in tropical and subtropical countries in Asia, Africa, Oceania and Latin America [1,2,3]. In Sub-Saharan Africa alone, an average of 250,000 snake envenomations per year is estimated, with more than 17,000 deaths [1], besides several morbidities, including local tissue damage, amputations and chronic disabilities [2].

*Bitis arietans*, also known as the “puff adder”, is considered a snake of great medical relevance. Its wide geographic distribution, close contact with human populations and domestic animals, large body size and venom rich in a variety of toxins explain its medical importance [4,5,6,7,8]. This snake is responsible for more accidents and deaths, in humans and domestic animals, than all other African snakes [4]. Like other snakes from the Viperidae family, *B. arietans* envenomation is characterized by local reactions, including pain, blistering, edema, neutrophilia and tissue damage, aggravated by the systemic reactions such as fever, hemostatic and cardiovascular disturbances [4,5,6,7,8]. As a result, many victims can die or be left with permanent disabilities and sequelae. The only specific treatment for snakebites is antivenom therapy [4]. However, antivenoms, besides being scarce and expensive in under-developed countries, are not able to neutralize all the effects of the envenomation, especially local ones [7,8,9]. Thus, studies on the *B. arietans* venom (BaV) mechanism of action are necessary for the development of effective complementary therapies. Despite the relevance of accidents involving the *B. arietans* snake, the mechanism underlying the envenomation remains poorly understood, especially regarding the involvement of inflammatory response, which can contribute to the worsening of local [10,11] and systemic reactions [6,7,8,9,10,11,12].

Inflammatory response is a complex mechanism which includes biochemical, vascular and cellular alterations in the site of tissue injury. These alterations lead to the activation of plasma molecules from coagulation and complement systems, accompanied by leukocytes’ activation and recruitment to inflammatory sites, which aim to control the harmful effect and restore homeostasis. Inflammation is essential for repairing damaged tissue, as there is no wound healing in the absence of an inflammatory process. However, if not properly controlled, it can trigger deleterious effects, such as tissue damage and loss of function of the inflamed area [13].

Several studies have shown that venoms from snakes of the Viperidae family can promote inflammation. The inflammatory-associated events involved in envenomation by snakes from the *Bothrops* genus have been extensively studied, and are characterized by increased vascular leakage, edema and accumulation of inflammatory cells at the injury site, such as polymorphonuclear leucocytes (PMNs). Cytokines, including IL-6, TNF-α and IL-1β, and chemokines, such as CCL-2, also known as MCP-1 also have their production increased [14,15,16,17]. These venoms also trigger the release of lipid mediators derived from membrane phospholipids, such as platelet-activating factor (PAF), and arachidonic acid (AA)-derived eicosanoids produced by cyclooxygenases 1/2 (COX1/2), such as prostaglandins (PGs) and thromboxane (TXs), and by 5-lipoxygenase (5-LO), giving rise to leukotrienes (LTs) [14,17,18,19]. These lipid mediators, which are produced by resident and inflammatory cells, are involved in several steps of the inflammatory response, including the development of pain, edema, chemotaxis and leukocyte activation, besides the resolution phase [20]. In this sense, it was already shown that pharmacological inhibition of eicosanoids can reduce the edema promoted by both *Bothrops asper* and *B. jararaca* venoms [21]. Furthermore, edema, myotoxicity and leukocyte recruitment induced by *B. jararaca* and *B. jararacussu* venoms can also be attenuated by pre-treatment with dexamethasone, an inhibitor of phospholipases A_2_ (PLA_2_), which are the enzymes responsible for membrane phospholipids hydrolysis and AA release [22]. Other works have also shown the beneficial role of drugs that block the lipid mediators’ synthesis in snake envenomations [23,24], indicating that the inflammatory events promoted by snake venoms are, at least in part, dependent on these molecules.

Despite the low number of studies on the characterization of the inflammatory process involved in the snake envenomation of the Viperinae subfamily, which includes the genus *Bitis*, a recent work was published showing the pro-inflammatory activity of the *Montivipera bornmuelleri* venom using a peritonitis mice model. After 6 h and 24 h of venom inoculation, the production of TNF-α, IFN-γ, IL-1β, IL-17, IL-4 and IL-10 were detected [25]. In another study, after treating U937 cells with *Vipera ammodytes ammodytes* venom, an increase in the expression of several inflammatory genes was observed, such as IL-1β and IL-8 [26]. The *Vipera berus* venom, another member of the Viperinae snake subfamily, caused edema, hemorrhage and myonecrosis in a model of inflammation induced in the cremaster muscle of mice [27].

However, despite extensive research into the inflammatory process associated with Viperidae envenomation, there are no studies on the inflammatory process induced by the *Bitis arietans* venom, in spite of the large number of accidents caused by this snake.

The main objective of this study was to evaluate the contribution of lipid-derived mediators in the inflammatory events induced by the *B. arietans* venom. For this, studies were carried out on the increasing vascular permeability, accumulation of leukocytes at the venom injection site, gene expression of inflammatory proteins, production of eicosanoids, production of cytokines and chemokines and induction of hemorrhage. The study was conducted by associating in vitro and in vivo experiments, including two genetically deficient mice: A- 5-lipoxygenase enzyme (5-LO^−/−^) and B- PAF receptor-deficient mice (PAFR^−/−^).

## 2. Results

### 2.1. Peritoneal Inflammation Induced by BaV

Initially, Evan’s Blue dye accumulated in the peritoneal cavities of 129 SvE mice i.p. injected with BaV (0.5 mg/kg) or saline as controls were examined. Figure 1A indicates that plasma extravasation is detected after 10 min of the venom injection attaining higher values after 30 min. Plasma extravasation was not detected in controls. In these periods, hemorrhage was not detected, while 4 h and 8 h after BaV injection, an erythrocytes accumulation was observed in the peritoneal cavities of envenomed animals, as evaluated by hemoglobin assays (Figure 1B). Next, the cell populations in the peritoneal cavities of groups of 129 SvE mice similarly challenged with BaV were examined. The peritoneal liquids were collected, and the cells were microscopically counted and morphologically analyzed considering nucleus conformation and the presence/absence of typical intracellular granulations. Accumulation of total leukocytes (TLs) was higher after 4 h of BaV injection (above 6 × 10^6^/mL) with a small, but not significant, decay after 8 h (Figure 1C). Figure 1C also shows that the mononuclear cell (MNs) amount was not different between the experimental and control groups, while polymorphonuclear leukocytes (PMNs) increased significantly in animals challenged with BaV, indicating that PMNs are the main cell population that contribute to the inflammatory exudate induced by the venom. Then, the functional state of the exudate cells was investigated. Gene-activation of important interleukins (IL-6 and IL-1β) and enzymes (COX-1, COX-2 and 5-LO) during intraperitoneal envenomation was analyzed and an increase of IL-6, IL-1β, COX-1 and COX-2 was observed, although, in a lesser extension, an increase of 5-LO also was observed (Figure 1D). The stimulation of another set of inflammation mediators, the eicosanoids, was next evaluated and Figure 1E shows the increase in the production of eicosanoids LTB_4_, LTC_4_, PGE_2_, and TXB_2_. Continuing with the characterization of the inflammatory response induced by BaV, the production of IL-6, IL-10, IFN-γ, TNF-α, IL-12p70, IL-1β and MCP-1 was evaluated. The cytokine/chemokine IL-6 and MCP-1, were local (Figure 1F) and systemic (Figure 1G) detected. IL-10, IFN-γ, TNF-α, IL-12p70 and IL-1β were not detected either locally or systemically (data not shown). The edema, cell infiltration and gene-activation induced by venom toxins created conditions for inflammation which were completed by the simultaneous induction of a hemorrhage. In general, these results confirm and expand information on the inflammatory process extensively described in humans or animals bitten by the *Bitis arietans* snake [4,5,6,7,8].

In view of the importance of lipid mediators in inflammatory events, their participation in the inflammation induced by BaV was investigated. To this purpose, new experiments were conducted under pharmacological intervention strategies and by the use of genetically deficient mice. The obtained results are described below.

### 2.2. Contribution of Leukotrienes to the Inflammatory Pathogenesis Induced by BaV

The 5-lipoxygenase (5-LO) enzyme is involved in leukotrienes synthesis [28]. Thus, the contribution of leukotrienes to BaV-induced inflammation was investigated by pre-treating the animals with MK-886, an inhibitor of the 5-lipoxygenase-activating enzyme, or by using mice genetically deficient in 5- lipoxygenase (5-LO^−/−^). First, C57BL/6 mice, a genetically different lineage, but using similar experimental conditions, were subcutaneously pre-treated with MK-886. Under this condition, a reduction of PMNs cells in the peritoneal cavity of mice challenged with BaV was observed, while the total leukocytes and MNs were not altered. (Figure 2A). Moreover, a significant reduction of eicosanoids LTB_4_, LTC_4_, and TXB_2_ (Figure 2B), besides local (Figure 2C) and systemic (Figure 2D) IL-6 and MCP-1, were observed. Unexpectedly, the pre-treatment with MK-886 increased the hemorrhage induced by BaV (Appendix A).

However, in 5-LO^−/−^ mice, the number of total and PMNs leukocytes was similarly increased, as in 129 SvE mice (Figure 3A). The local production of IL-6 and MCP-1 was also similarly increased as in 129 SvE (Figure 3B), while the systemic concentrations of IL-6 and MCP-1 in 5-LO^−/−^ mice were higher than in the 129 SvE mice (Figure 3C). Interestingly, the increase of plasma proteins in the peritoneal cavity induced by BaV was higher in 5-LO^−/−^ in comparison with 129 SvE mice (Appendix A). In the same way, the hemorrhage after 4 h of BaV injection, as evaluated by hemoglobin quantification, was higher in 5-LO^−/−^ mice in comparison with 129 SvE mice (Appendix A). Aiming to validate the genetically deficient mice model, some aspects were evaluated. As expected, the enzyme 5-lipoxigenase expression was not detected in 5-LO^−/−^ mice after BaV injection (Appendix A) and the 5-LO downstream products, LTB_4_ and LTC_4_, were extensively reduced in 5-LO^−/−^ mice in comparison with 129 SvE mice; while the productions of PGE_2_ and TXB_2_ were not changed (Appendix A).

### 2.3. Contribution of PAFR to Inflammation Induced by the Bitis arietans Snake Venom (BaV)

The PAFR contribution to the inflammatory process induced by BaV was analyzed in C57BL/6 mice pre-treated with WEB-2086, a PAFR receptor antagonist, or in mice genetically deficient in PAF receptors (PAFR^−/−^). The accumulation of total, MNs and PMNs leukocytes in the intraperitoneal cavity was slightly reduced under WEB-2086 pre-treatment (Figure 4A). Moreover, WEB-2086 treatment did not change the production of LTB_4_ and PGE_2_, but effectively induced a decrease of LTC_4_ and TXB_2_ (Figure 4B). The local (Figure 4C) and systemic (Figure 4D) production of IL-6 was significantly reduced with WEB-2086 treatment, whereas the reduction of MCP-1 was less significant.

The intraperitoneal cavity hemorrhage, as evaluated by hemoglobin quantification, induced by BaV, was similar after WEB-2086 treatment (Appendix A) while it was attenuated in PAFR^−/−^ mice (Appendix A). In PAFR^−/−^ mice the reduction of total and PMNs leucocytes was significant in comparison with C57BL/6 mice (Figure 5A). In agreement, it was observed that there was a decrease in local (Figure 5B) and systemic (Figure 5C) IL-6 and MCP-1 production in PAFR^−/−^ mice.

### 2.4. Prostanoids Are Involved in the Inflammation Induced by BaV

The pre-treatment of C57BL/6 mice with indomethacin before BaV injection reduced the migration of total and PMN leukocytes into the peritoneal cavity of mice (Figure 6A). Indomethacin also reduced the production of LTC_4_ and TXB_2_, but not of PGE_2_, while the production of LTB_4_ was increased (Figure 6B). The pre-treatment with indomethacin was able to reduce local (Figure 6C) and systemic (Figure 6D) IL-6 production, but did not interfere in the synthesis of MCP-1. Still, as observed with the MK-886 treatment, mice pre-treated with indomethacin showed an increase in hemorrhage on the peritoneal cavity (Appendix A). This set of results shows that indomethacin pre-treatment induced a reduction of inflammatory cells and mediators in the peritoneal cavity of mice injected with BaV, and that this reduction occurred in the absence of an inhibition of PGE_2_ production in 30 min (Figure 6B) as well as in 4 h (Appendix A) after BaV injection.

However, an efficient and abrupt inhibition of TXB_2_ production, the main product of the COX-1 pathway [29], led us to investigate the participation of the COX-1 pathway downstream products in inflammation induced by BaV. To this purpose, mice were pre-treated with SC-560, a selective COX-1 inhibitor. Under this condition, there was a decrease in total and PMN leukocytes migration (Figure 7A), while no reduction was observed in the local (Figure 7B) or systemic (Figure 7C) production of IL-6 and MCP-1. The treatment with SC-560 did not change the COX-1 expression after 4 h of BaV injection, whereas the expression of COX-2 was not detected in this period (Figure 7D). In the same way after WEB-2086 treatment, SC-560 did not interfere with the hemorrhage caused by BaV in the mice peritoneal cavities (Appendix A).

## 3. Discussion

The use of the *Bitis arietans* venom as an inflammation inducer, the peritoneal cavity as the anatomical site, and the 129 SvE and C57BL/6 mice as animal models in this study were all motivated by the following: (a) *B. arietans* is a snake widely distributed throughout Sub-Saharan Africa, and Morocco and West Arabia [1,2,3,30,31]; (b) the envenomation causes inflammation, and severe local and systemic symptoms, which may lead to death [4]; (c) the peritoneal cavity was chosen for analysis of BaV-inducing inflammation due to its viability as an experimental model; (d) the use of 129 SvE mice limits the genetic influence, while the use of C57BL/6 generalizes the obtained results; (e) the total venom was used to get an overview of the whole process.

In addition, this study used pharmacological inhibitors, and mice deficient in strategic genes, which allowed for the evaluation of important data on BaV pro-inflammatory activity. The obtained results showed local leukocyte accumulation in the peritoneal cavity, increased expression of genes encoding IL-6, IL-1β, COX-1, COX-2 and 5-LO, production of important inflammatory cytokines and chemokines, such as IL-6 and MCP-1. The eicosanoids LTB_4_, LTC_4_, PGE_2_ and TXB_2_ also had their production increased. The role of PAFR in the peritoneal cavity cells was also evaluated. This inflammatory process leads to local induction of hemorrhage. The venom-induced increase in vascular permeability was demonstrated in 129 SvE and 5-LO^−/−^ mice early after 10 min. However, C57BL/6 and PAFR^−/−^ lineages, with the same genetic background, developed early hemorrhage caused by i.p. inoculation of BaV (0.5 mg/kg). Therefore, it was not possible to evaluate the increase in venom-induced vascular permeability in this experimental model, but it was shown that the C57BL-6 mice were more sensitive in the development of BaV-induced hemorrhage.

Mice genetically deficient of 5-lipoxygenase (5-LO^−/−^) showed unexpected and different results when were pre-treated with the 5-LO inhibitor, MK-886. Instead of the expected reduction of evaluated inflammatory cells, after MK-886 treatment, no changes in 5-LO^−/−^ mice were observed. Moreover, 5-LO^−/−^ mice showed a systemic increase of IL-6 and MCP-1. This genetically deficient lineage had been already used in a previous study with the *Tityus serrulatus* scorpion’s venom [32], in which the animals developed attenuated inflammatory response. This validates the used experimental model, and shows the complexity of interactions between different toxins from animal venoms and the immune response.

However, the inflammation induced by BaV is followed by hemorrhage, which may indicate that other plasma components, including platelets, reached the peritoneal cavity, increasing the complexity of the process. In this sense, it is well known that platelets mostly express COX-1 [33] and 12-LO [34]. The 12-LO expression may explain the reminiscent production of leukotrienes after MK-886 treatment, but does not explain the divergence between results from mice deficient of 5-LO^−/−^ from mice pre-treated with the inhibitor. Both showed inhibition of 5-LO activity and a possible contribution of the platelets that leaked into the peritoneal cavity as a result of hemorrhage. Pharmacological interventions are complex, and can interfere with more than one pathway. Although MK-886 was originally described as a selective FLAP inhibitor, this compound also inhibits COX-1 and platelet aggregation [35]. In this scenario, the TXB_2_ reduction, main product of COX-1 pathway [29], in mice treated with MK-886—but not in 5-LO^−/−^ mice—may indicate the importance of the COX-1 pathway in BaV induced inflammation. MK-886 can also inhibit PPAR-α [36], which is associated with the complement component 3 (C3) expression [37,38], possibly indicating the participation of this system in these inflammatory events, as already described in studies with other venoms [39,40,41]. Summing up, this set of results indicated that only the reduction of leukotrienes—as observed in both experimental conditions described here—is not sufficient to attenuate the inflammation induced by BaV.

This work also showed the decrease in the number of inflammatory molecules, but not in PMN cell migration, after WEB-2086 treatment, while in PAFR^−/−^ mice all parameters evaluated were attenuated. The pharmacological inhibition protocols, including dose, time and route of treatment, are directly linked to the efficiency and duration of PAFR inhibition [19]. On the other hand, the use of PAFR^−/−^ mice demonstrated the participation of PAFR in inflammation promoted by BaV. Finally, the pharmacological inhibition of COX-1/2 metabolites using indomethacin and the specific COX-1 inhibitor (SC-560) demonstrated a) the role of prostanoids in the inflammatory process promoted by BaV, including TXB_2_, but not PGE_2_ to the PMNs leukocytes migration; b) the production of IL-6, but not of MCP-1. These observations are in accordance with published results, where the treatment with indomethacin reduced the number of PMN cells in local inflammation induced by the *B. jararacussu* venom [23]. In other studies, indomethacin inhibited the expression of the IL-6 gene, but not of TNF-α [42].

Interestingly, an increase in erythrocytes was observed in the peritoneal cavity of mice after interventions with MK-886 and indomethacin, when compared with mice treated with BaV alone. However, a decrease in the total and leukocyte migration of PMNs was also observed with both pharmacological treatments, indicating that the endothelial wall’s rupture linked to hemorrhage is not enough for leukocyte extravasation.

The experimental models here used, and the results obtained, may be useful to understand the role of inflammation in BaV envenomation, hence potentially improving the victim’s treatment, in addition to indicating the possible toxins responsible for these symptoms. In this sense, Kn-Ba, a serine protease with fibrinolytic and kinin-releasing activity, was isolated from the *Bitis arietans* venom [43] and it is also related to the inflammatory process, including the activation of the inflammasome complex and the release of active IL -1β. For this, new studies are being finalized.

## 4. Conclusions

This study shows that BaV induces an acute in vivo inflammatory response which was characterized by the increase of vascular leakage, the accumulation of PMNs cells and the production of eicosanoids, IL-6 and MCP-1, besides the participation of PAFR. This inflammatory process is associated with an influx of other blood components, such as platelets, due to hemorrhage. Through pharmacological interventions with anti-inflammatory drugs and using knock-out mice, it was possible to conclude that these inflammatory events promoted by the *B. arietans* venom are partially dependent on lipid mediators (Figure 8).

## 5. Materials and Methods

### 5.1. Snake Venom

Lyophilized *B. arietans* venom was purchased from Venom Supplies, Tanunda, Australia. The sample used was made up of venoms collected from healthy adult snakes (60 cm) of both genders, originally captured in South Africa, and maintained in captivity. The venom was diluted in pyrogen-free saline 0.15 NaCl, and submitted to protein quantification and endotoxin contents determination.

### 5.2. Protein Contents Determination

The protein contents were determined by the bicinchoninic acid (BCA) method [44] using the commercial kit Pierce BCA Protein assay (Pierce Biotechnology, Rockford, IL, USA). A standard curve was constructed with increasing concentrations of bovine serum albumin (BSA, Aldrich, MO, USA) diluted in PBS buffer (8.1 mM Na_2_HPO_4_; 1.5 mM KH_2_PO_4_; 137 mM NaCl; 2,7 mM KCL; pH 7.4). After protein determination, venom samples were stored at −80 °C.

### 5.3. Endotoxin Contents Determination

BaV samples (20 µg/mL) were analyzed by the Semi-Quantitative LAL test (*Limulus amebocyte* lysate) using the PYROGENT TM Gel clot LAL Assays kit (Lonza, MD, USA), according to instructions. Estimated endotoxin concentrations detected in the venom samples were calculated using *Escherichia coli* (0.125 UE/mL) as standard. The acceptable endotoxin concentration in 1.0 µg of venom is <0.1 UE [45].

### 5.4. Animals

Mice (males and females, 20–22 g) of four distinct lineages were used in the experiments. C57BL/6, 129 SvE and 129 SvE deficient in the 5-lipoxigenase enzyme (5-LO^−/−^) were provided by the Central Animal Facility, (CEBIOT) from São Paulo University (USP). C57BL/6 PAF receptor deficient (PAFR^−/−^) were a gift from Prof. Takao Shimizu, Tokyo University, Japan. As the PAFR^−/−^ and 5-LO^−/−^ mice belong to C57BL/6 and 129 SvE inbred background, respectively, obtained results were compared with the corresponding lineages. All animal experiments were approved by the institutional animal commit (CEUA ICB; N°: 4669091117), on 12 May 2017. According to CONCEA recommendations, the animals were maintained under healthy conditions, receiving water and food ad libitum.

### 5.5. BaV Injection and Blood and Peritoneal Exudate Collection

Mice were intraperitoneally (i.p.) injected with myotoxic and a non-lethal dose of BaV (0.5 mg/kg) [46,47] diluted in 200 µL of 0.15 NaCl, pyrogen-free saline. Control mice were injected, also i.p., with an equivalent volume of pyrogen-free 0.15 NaCl saline. After 30 min, 4 h or 8 h, as indicated in previously articles [14,17,48], the animals were anesthetized by isoflurane inhalation, and euthanized with CO_2_. For systemic quantification of inflammatory molecules, blood was collected by cardiac puncture and immediately transferred to tubes containing anticoagulant (Trisodium citrate dihydrate; 129 mmol) in the ratio of 1 part of anticoagulant to 9 parts of blood (1:9; *v*/*v*). After gentle shaking, the blood was 5000× *g* centrifuged at 4 °C for 10 min and the obtained plasma was stored at −80 °C. The peritoneal exudates were collected for local quantification of inflammatory molecules and hemoglobin. Thus, samples of 3 mL of ice-cold PBS (pH 7.4) were injected into the peritoneal cavity and, after a gentle massage for 1 min, the cell-rich fluid was collected and centrifuged at 500× *g* for 5 min at 4 °C. The supernatants were collected and stored at –80 °C for further quantification of cytokines, chemokines and lipid mediators. The pelleted cells were diluted with 200 µL of ice-cold PBS and cell suspensions were used to identify and count leukocytes populations, and to quantify hemoglobin.

### 5.6. Peritoneal Exudate Analysis

#### 5.6.1. Total and Differential Cell Counting

The pelleted cells obtained from centrifuged exudates were diluted in 200 µL of ice-cold pH 7.4 PBS. Aliquots were diluted in Turk’s solution (1:20; *v*/*v*) for total cell counting in a Neubauer chamber under light microscopy. For cell types differential scoring, exudate smeared cells (5 × 10^4^ cells) were prepared on a glass microscopic slide, and stained with Instant Prov (Dyslab, Paraná, Brazil). A minimal of 100 cells was randomly counted and mononuclear and polymorphonuclear cells were identified. The percentage of each cell type was adjusted to the total number/mL.

#### 5.6.2. Peritoneal Hemorrhage

The peritoneal erythrocytes influx was used to evaluate the BaV-induced hemorrhage. Samples from the pelleted cells were diluted 1:60 in deionized water and incubated for ten minutes at room temperature. After hemolysis, released hemoglobin was titrated at λ 415 nm in a plate spectrophotometer (ELX 800, Biotek Instruments, Winooski, VT, USA). In parallel, a standard curve was prepared with increasing concentrations of purified bovine hemoglobin (Sigma Aldrich, St. Louis, MO, USA) diluted in pH 7.4 PBS: 0 g/dL, 0.0039 g/dL, 0.0078 g/dL, 0.0156 g/dL, 0. 312 g/dL, 0.05 g/dL, 0.1 g/dL, 0.2 g/dL, 0.3 g/dL and 0.4 g/dL.

### 5.7. RNA Analysis of Peritoneal Exudate Cells

Individual samples from pelleted peritoneal cell exudates diluted in ice-cold PBS (160 µL/animal) were mixed with lysis buffer (ACK lysing buffer, Gibco, Invitrogen Corp, Carlsbad, CA, USA), and left in an ice bath for 3 min. After the addition of 2.0 mL of ice-cold pH 7.4 PBS, the mixture was shaken and centrifuged for 4 min, at 4 °C, at 250× *g*. The supernatants were carefully discarded, and 250 µL of Trizol were added into precipitates, and the samples stored at –80 °C. Total RNA was extracted by passing the supernatants through a “Direct-zol^TM^ RNA MiniPrep” (Zymo Research, CA, USA). The concentration of the extracted RNA was determined in a spectrophotometer (VersaMax^TM^, Molecular Devices, San Jose, CA, USA) and the resulting concentration was adjusted to synthesize the corresponding cDNA. cDNAs were synthesized using the RT-qPCR kit Verso cDNA Synthesis (Thermo Scientific, Waltham, MA, USA) using the following specific primers: Il-6 (F: 5′-TCC TTA GCC ACT CCT TCT GT-3′; R: 5′-AGC CAGAGT CCT TCA GAG A-3′), Cox-1 (F: 5′-TTC AAC ACA CTC TAT CAC TGG C-3′; R: 5′-AGA AGC GTT TGC GGT ACT CAT-3′), Cox-2 (F: 5′-CTC CCT GAA GCC GTA CAC AT-3′; 5′-ATG GTG CTC CAA GCT CTA CC-3′), acquire from of Exxtend (Exxtend Solução em Oligos, SP, Brazil), as well as for Il-1β (F: 5′-CTC TTG TTG ATG TGC TGC TG-3′; R: 5′-GAC CTG TTC TTT GAA GTT GAC G-3′), Alox5 (F: 5′-CCA GTC GTA CTT TGAATC CGT-3′; R: 5′-CCA TCT GCC TGC TAT ATA AGA ACC) and Hprt (F: 5′-AGC AGG TCA GCA AAG AAC T-3′; R: 5′-CCT CAT GGA CTG ATT ATG GACA-3′) used as control, from Integrated DNA Technologies (IDT, Coralville, IA, USA). The SYBR Green Fast (Applied Biosystems, Life Technologies, CA, USA, in the StepOne Plus Real-Time PCR system (Thermo Scientific, Waltham, MA, EUA), was used. Relative expressions of desired genes were normalized by the simultaneous expression of the control gene (Hprt). The analysis was performed by the method 2^−ΔΔCt^, as indicated by [49].

### 5.8. Expression of the COX-1/2 Proteins by Western Blotting

The pellet of peritoneal exudate cell samples was suspended in lysing buffer (ACK lysing buffer, Gibco, Invitrogen Corp, Carlsbad, CA, USA). After cell lysis, the supernatants were carefully removed and 40 µL of buffer containing β-mercaptoethanol for each 1 × 10^6^ cell samples were added. After heating for 10 min at 100 °C, the samples were stored at −20 °C. The protein expression in the supernatants was analyzed by electrophoresis [50] followed by Western Blotting [51]. Raw 264.7 cells were treated (24 h) with LPS (100 µg/mL) and were also reduced with β-mercaptoethanol, and used as COX-2 positive controls [52]. Samples of 20 µg of cell extracts, both experimental and control, were deposited in a corresponding gel well. After electrophoresis, the protein bands were transferred to polyvinylidene difluoride membrane (PVDF, GE Healthcare, Chicago, IL, USA), in a semi-dried system (Trans-Blot SD Semi-Dry Transfer Cell, Bio-Rad, Hercules, CA, USA) for 35 min at 15 V. Non-specific sites were blocked with 5% non-fat milk diluted in TBST (150 mM NaCl, 29 mM Tris, 0. 5% Tween 20, at pH 7.4) for 1 h at room temperature. After washing with TBST, the membranes were incubated overnight at 4 °C with anti-COX-1 and COX-2 monoclonal antibodies diluted 1:500 (Cayman Chemical, Ann Arbor, MI, USA). As a control, similar membranes were incubated under the same conditions with anti-β-actin diluted 1:2000 (Cell Signaling, Danvers, MA, USA). After washing with TBST, the protein bands were combined with murine or rabbit anti-IgG peroxidase conjugated secondary antibodies, and protein bands expressing positive interactions were identified by a chemiluminescence reaction using the ECL substrate (Thermo Scientific, MA, USA). The band intensities were determined by densitometry using the analytical program Alpha DigiDoc 1000 v3.2 Beta (Alpha Innotech Corp., San Leandro, CA, USA) and normalized with β-actin.

### 5.9. Quantification of Lipid Inflammatory Mediators in Peritoneal Exudates

The concentration of lipid inflammatory mediators, LTB_4_, LTC_4_, PGE_2_ and TXB_2_, a stable TXA_2_ metabolite, in the peritoneal exudates of mice injected with BaV, was quantified by ELISA using specific kits (EIA, Cayman Chemical, Ann Arbor, MI, USA). Absorbance was determined in a spectrophotometer at λ 405/420 nm (VersaMax™, Molecular Devices, CA, USA). Eicosanoid concentrations were calculated based on the kit standard curve, using as inferior limits >13 pg/mL (LTB_4_, and TXB_2_), >10 pg/mL (LTC_4_) and >15 pg/mL (PGE_2_).

### 5.10. Quantification of Cytokines and Chemokines in Peritoneal Exudates

The cytokines and chemokines profiles induced by the BaV were evaluated using peritoneal exudates’ free cells fluid. For quantification of inflammatory chemokines TNF-α, IL-6, MCP-1, IFN-γ, IL-10 and IL-12p70, the CBA Mouse Inflammatory Cytokines (BD, Bioscience CA USA) kit was used. The experiments were conducted in a flow cytometer (FACS Canto II, Becton Dickinson, Franklin Lakes, NJ, USA), and data was analyzed by the software FCS Diva, version 6.1.3. The lower detection limits considered were: >7.3 pg/mL (TNF-α); >5 pg/mL (IL-6); >52.7 pg/mL (MCP-1); >2.5 pg/mL (IFN-γ); 17.5 pg/mL (IL-10); and 10.7 pg/mL (IL-12p70). The cytokine IL-1β was quantified using the ELISA kit Mouse IL-1β ELISA Set (BD Biosciences, CA, USA). The absorbance was read on a plate spectrophotometer (ELX 800, Biotek Instruments, Winoosk, VT, USA) at λ 450 nm and the result was interpolated on the standard curve. The internal limit considered was >15 pg/mL.

### 5.11. Quantification of IL-6 and MCP-1 in the Murine Plasmas

The IL-6 and MCP-1 were titrated in mouse plasma using the kit BD OptEIA™ Set Mouse (BD Bioscience, San Jose, CA, USA), according to the manufacturer’s instructions. The absorbance was determined in a plate spectrophotometer, optical density λ 450 nm (ELX 800, Biotek Instruments, Winoosk, VT, USA), and the values were obtained from a standard curve. The kit detection ranges between 10 and 1000 pg/mL.

### 5.12. Increasing Vascular Permeability Evaluation

The increase in vascular permeability induced by BaV was evaluated by the Evan’s Blue dye diffusion into the peritoneal cavity [14,53]. Groups of mice were anesthetized by isoflurane and injected through the retro-orbital plexus [54] with 100 µL sterile LPS-free of Evan’s Blue solution. Each animal received 20 mg/kg of the dye. Next, the animals were injected i.p. with BaV (0.5 mg/kg). Control mice were injected only with sterile saline. After 10 and 30 min, mice were anesthetized by isoflurane and euthanized under a CO_2_ atmosphere, and 3 mL of pH 7.4 sterile PBS were injected into the peritoneal cavities. After gentle massage, 2.0 mL of peritoneal exudates were collected and centrifuged at 500× *g*, 5 min; supernatants were also collected. The absorbance of Evan’s Blue in the supernatants was measured at λ 625 nm (ELX 800, Biotek Instruments, Winoosk, VT, USA) and the values interpolated in the standard curve (0.3–50 µg/mL).

### 5.13. Pharmacological Interventions

The following anti-inflammatory drugs in indicated doses were used: indomethacin (10 mg/kg) [23], SC-560 (5 mg/kg) [55], MK-886 (5 mg/kg) [56], WEB-2086 (5 mg/kg) [19], inhibitors of COX-1/2, COX-1, 5-LO–activating protein (FLAP) and PAF receptor antagonist, respectively. The inhibitors (Cayman Chemical, USA) were diluted in DMSO and stored at –80 °C. A sterile 0.15 M NaCl solution containing the inhibitors, not exceeding the final volume of 150 µL per mouse, was prepared at the time of use. Experimental C57BL/6 groups were subcutaneously (s.c.) injected at the dorsal regions with the indicated anti-inflammatory drugs. Control groups were injected with the drugs and the solvents used were emulsified in sterile 0.15 M NaCl. After 1 h, the animals were i.p. challenged with 0.5 mg/kg of BaV. After 30 min and 4 h, peritoneal exudates were collected from groups of mice, and used for differential leukocyte counting and lipid mediator, cytokine and chemokine quantification. In parallel, the hemorrhage was also evaluated through hemoglobin quantification.

### 5.14. Statistical Analysis

Data are presented as means ± SD and analyzed by Graph Pad Prism, version 6.0 for Windows (Software, San Diego, CA, USA, 2017). The Student’s *t* test was used to compare the results of two groups. One-Way ANOVA test and multiple comparisons by Tukey HSD were used for comparisons of one variable in more than two groups. For comparisons of two or more variables, Two-Way ANOVA followed by Tukey HSD were used. For all tests, the values *p* < 0.05 were considered significant.

## Figures and Tables

**Figure 1 toxins-12-00594-f001:**
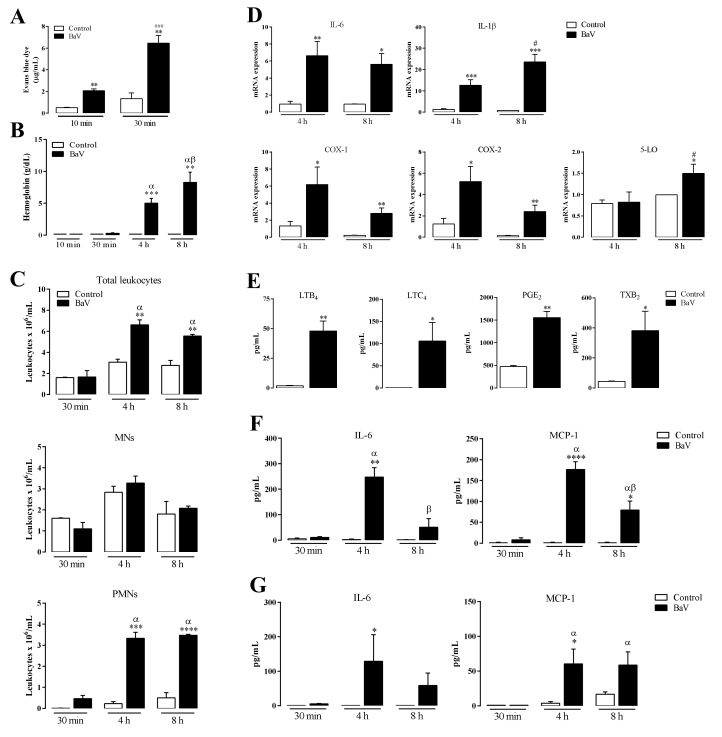
Characterization of inflammatory response induced in 129 SvE mice by BaV. Mice were i.p. injected with BaV (0.5 mg/kg) or sterile saline as a control. After determined periods, as indicated, peritoneal exudate was harvest to evaluated: (**A**) the increase of vascular permeability by Evan’s Blue dye extravasation; (**B**) hemorrhage by hemoglobin quantification; (**C**) accumulation of total, mononuclear (MNs) and polymorphonuclear leukocytes (PMNs); (**D**) gene-activation of IL-6, an IL-1β, and the enzymes COX-1, COX-2 and 5-LO; (**E**) the production of eicosanoids LTB_4_, LTC_4_, PGE_2_ and TXB_2_, after 30 min of BaV inoculation; and (**F**) the production of local IL-6 and MCP-1. (**G**) Plasma samples were used to detect the production of systemic IL-6 and MCP-1 after 4 h of BaV inoculation. (*) Differences between BaV and control; (#) differences between two periods after BaV inoculation; (α) difference of 4 h and 8 h after BaV inoculation compared to 30 min; (β) difference of 8 h after BaV inoculation compared to 4 h. Results were expressed as mean ± SD (4–5 mice) of three reproducible assays. (* or #) *p* < 0.05, (**) *p* < 0.01, (*** or ###) *p* < 0.001 and (****) *p* < 0.0001.

**Figure 2 toxins-12-00594-f002:**
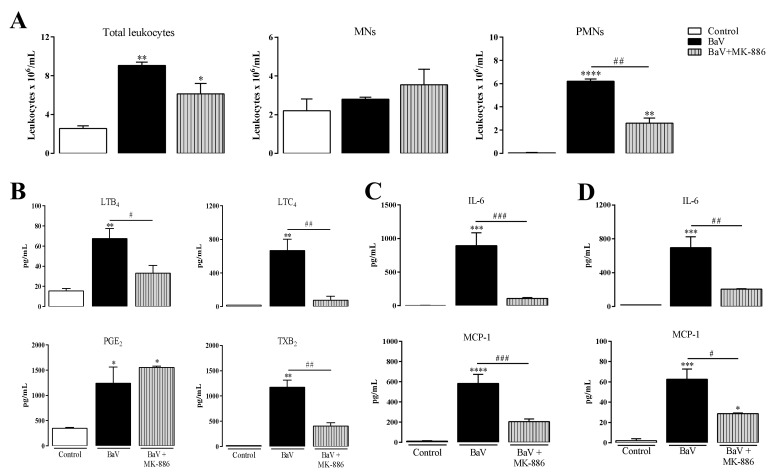
MK-886 treatment decreases inflammation induced by BaV. One hour before i.p. BaV inoculation (0.5 mg/kg), C57BL/6 mice were s.c. treated with MK-886 (5 mg/kg). After selected periods of BaV inoculation, peritoneal exudate was harvest to evaluated: (**A**) accumulation of total, mononuclear (MNs) and polymorphonuclear leukocytes (PMNs) after 4 h; (**B**) the production of eicosanoids LTB_4_, LTC_4_, PGE_2_ and TXB_2_, after 30 min; and (**C**) the production of local IL-6 and MCP-1 after 4 h. (**D**) Plasma samples were used to detect the production of systemic IL-6 and MCP-1 after 4 h of BaV inoculation. (*) Differences between BaV and control; (#) differences between mice pre-treated or not with MK-886. Results were expressed as mean ± SD (4–5 mice) of three reproducible assays. (* or #) *p* < 0.05, (** or ##) *p* < 0.01, (*** or ###) *p* < 0.001 and (****) *p* < 0.0001.

**Figure 3 toxins-12-00594-f003:**
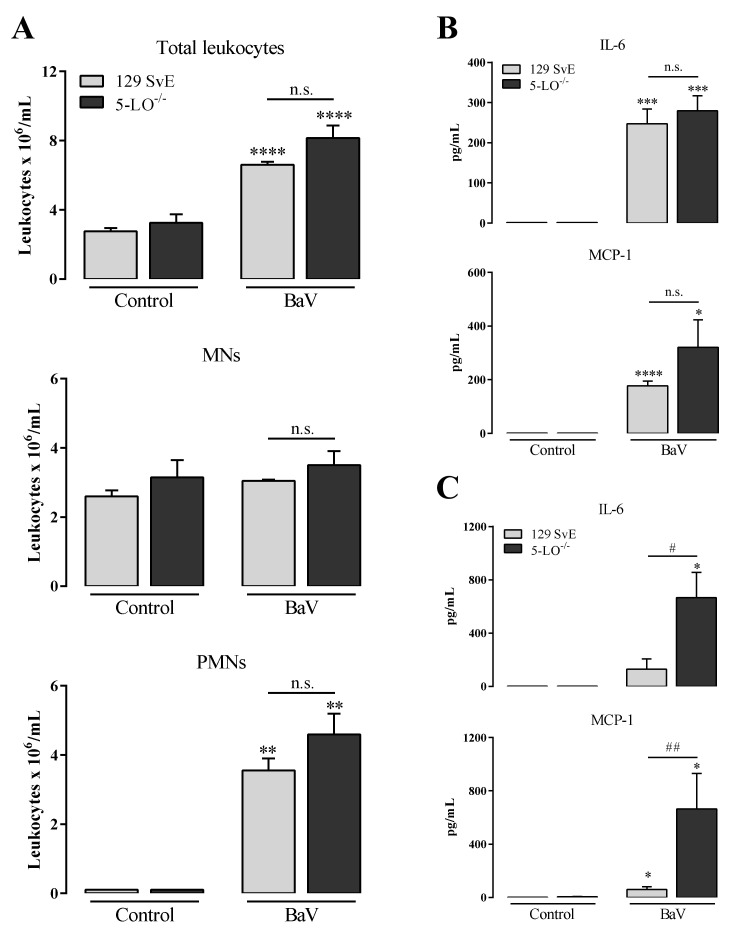
Inflammation induced by BaV is not attenuate in mice genetically deficient in the 5-LO enzyme. BaV (0.5 mg/kg) was i.p. inoculated in mice genetically deficient in the 5-LO enzyme (5-LO^−/−^) or in 129 SvE mice. After 4 h, peritoneal exudate was harvested and evaluated: (**A**) accumulation of total, mononuclear (MNs) and polymorphonuclear leukocytes (PMNs); and (**B**) the production of local IL-6 and MCP-1 after 4 h. (**C**) Plasma samples were used to detect the production of systemic IL-6 and MCP-1, also after 4 h of BaV inoculation. (*) Differences between BaV and control; (#) differences between 5-LO^−/−^ and 129 SvE mice. Results were expressed as mean ± SD (4–5 mice) of three reproducible assays. (* or #) *p* < 0.05, (** or ##) *p* < 0.01, (***) *p* < 0.001 and (****) *p* < 0.0001. n.s.: not significant.

**Figure 4 toxins-12-00594-f004:**
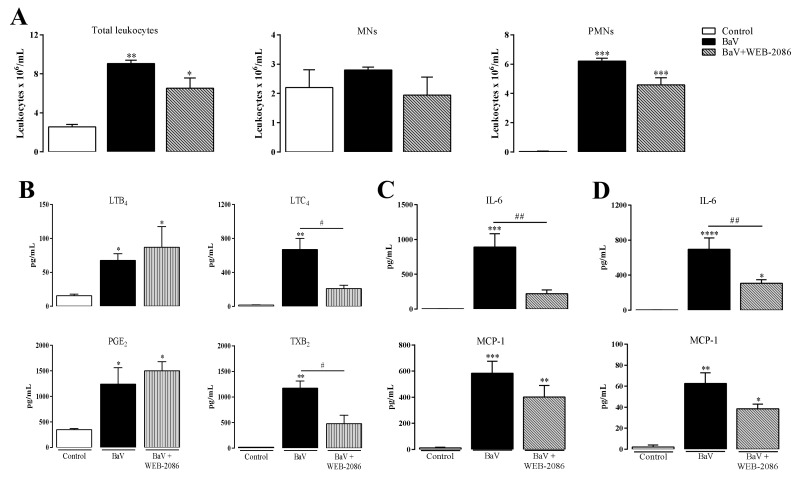
WEB-2086 treatment decreases the production of LTC_4_, TXB_2_ and IL-6 induced by BaV. One hour before i.p. BaV inoculation (0.5 mg/kg), C57BL/6 mice were s.c. treated with WEB-2086 (5 mg/kg). After selected periods of BaV inoculation, peritoneal exudate was harvested to evaluate: (**A**) accumulation of total, mononuclear (MNs) and polymorphonuclear leukocytes (PMNs) after 4 h; (**B**) the production of eicosanoids LTB_4_, LTC_4_, PGE_2_ and TXB_2_, after 30 min; and (**C**) the production of local IL-6 and MCP-1 after 4 h. (**D**) Plasma samples were used to detect the production of systemic IL-6 and MCP-1 after 4 h of BaV inoculation. (*) Differences between BaV and control; (#) differences between mice pre-treated or not with WEB-2086. Results were expressed as mean ± SD (4–5 mice) of three reproducible assays. (* or #) *p* < 0.05, (** or ##) *p* < 0.01, (***) *p* < 0.001 and (****) *p* < 0.0001.

**Figure 5 toxins-12-00594-f005:**
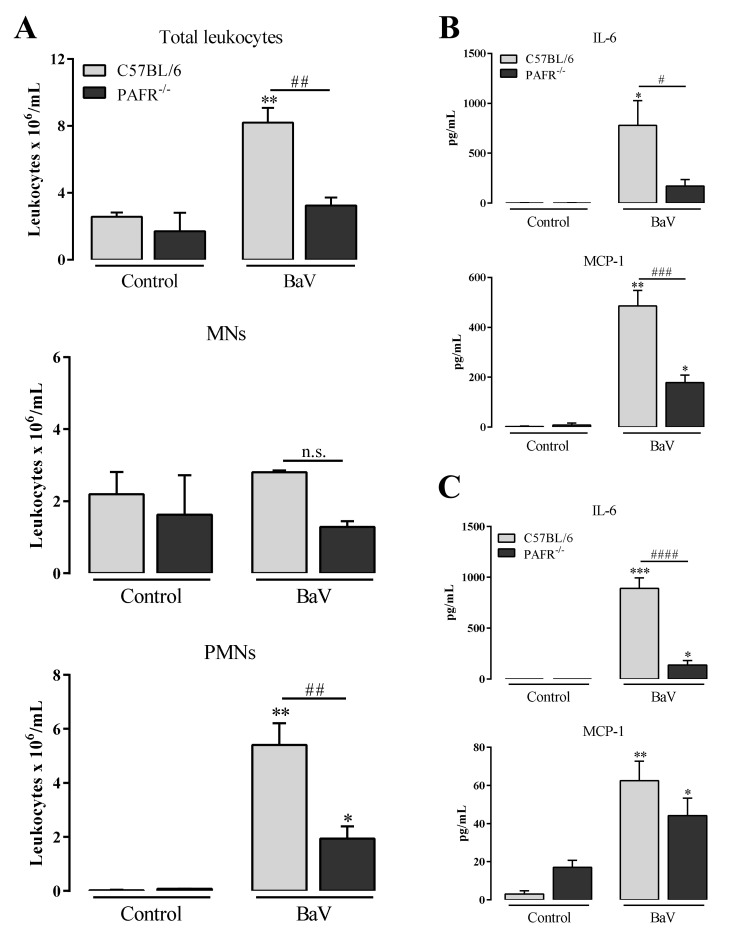
Mice genetically deficient in PAF receptors developed milder acute inflammation induced by BaV. BaV (0.5 mg/kg) was i.p. inoculated in mice genetically deficient in PAF receptors (PAFR^−/−^) or in C57BL/6 mice. After 4 h, peritoneal exudate was harvested to evaluate: (**A**) accumulation of total, mononuclear (MNs) and polymorphonuclear leukocytes (PMNs); and (**B**) the production of local IL-6 and MCP-1 after 4 h. (**C**) Plasma samples were used to detect the production of systemic IL-6 and MCP-1, also after 4 h of BaV inoculation. (*) Differences between BaV and control; (#) Differences between PAFR^−/−^ and C57BL/6 mice. Results were expressed as mean ± SD (4–5 mice) of three reproducible assays. (* or #) *p* < 0.05, (** or ##) *p* < 0.01, (*** or ###) *p* < 0.001 and (####) *p* < 0.0001. n.s.: not significant.

**Figure 6 toxins-12-00594-f006:**
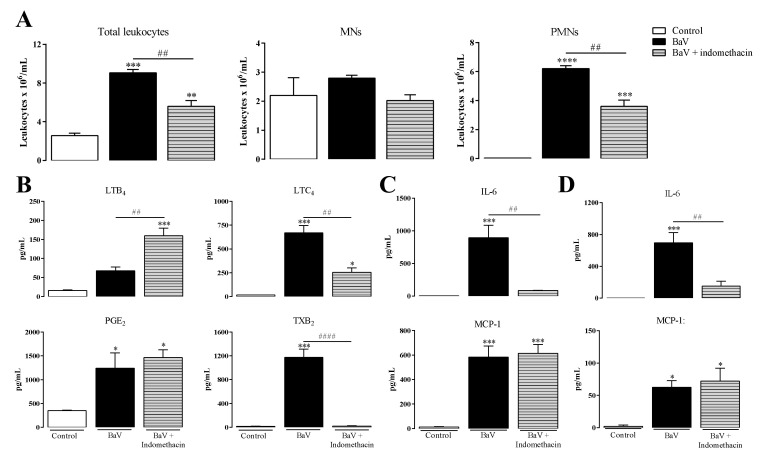
Inflammation promoted by BaV is partially dependent on COX-1/2 pathways products. One hour before i.p. BaV inoculation (0.5 mg/kg), C57BL/6 mice were s.c. treated with indomethacin (10 mg/kg). After selected periods of BaV inoculation, peritoneal exudate was harvested to evaluate: (**A**) accumulation of total, mononuclear (MNs) and polymorphonuclear leukocytes (PMNs) after 4 h; (**B**) the production of eicosanoids LTB_4_, LTC_4_, PGE_2_ and TXB_2_, after 30 min; and (**C**) the production of local IL-6 and MCP-1 after 4 h. (**D**) Plasma samples were used to detect the production of systemic IL-6 and MCP-1 after 4 h of BaV inoculation. (*) Differences between BaV and control; (#) differences between mice pre-treated or not with indomethacin. Results were expressed as mean ± SD (4–5 mice) of three reproducible assays. (*) *p* < 0.05, (** or ##) *p* < 0.01, (***) *p* < 0.001 and (**** or ####) *p* < 0.0001.

**Figure 7 toxins-12-00594-f007:**
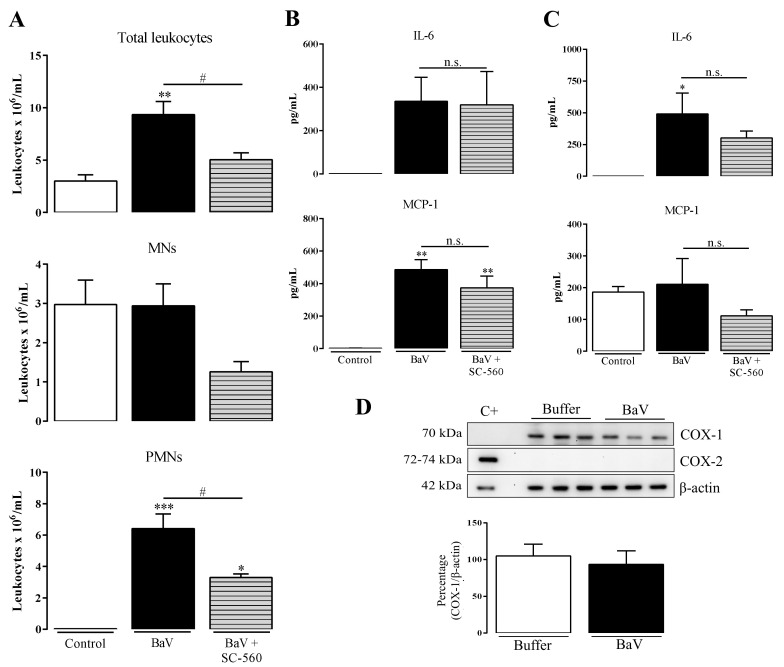
COX-1 pathway products are involved in the influx of PMNs cells into the site of inflammation induced by BaV. One hour before i.p. BaV inoculation (0.5 mg/kg), C57BL/6 mice were s.c. treated with SC-560 (5 mg/kg). After 4 h of BaV inoculation, peritoneal exudate was harvested to be evaluated: (**A**) accumulation of total, mononuclear (MNs) and polymorphonuclear leukocytes (PMNs); and (**B**) the production of local IL-6 and MCP-1. (**C**) Plasma samples were used to detect the production of systemic IL-6 and MCP-1 after 4 h of BaV inoculation. (**D**) Peritoneal exudate cells treated with lysis buffer were used to evaluate the expression of COX-1 and COX-2 by Western Blotting. Raw cells treated with LPS (24 h) were used as a positive control (C+) of COX-2 expression. (*) Differences between BaV and control; (#) differences between mice pre-treated or not with SC-560. Results were expressed as mean ± SD (4–5 mice) of three reproducible assays. (* or #) *p* < 0.05, (**) *p* < 0.01 and (***) *p* < 0.001. n.s.: not significant.

**Figure 8 toxins-12-00594-f008:**
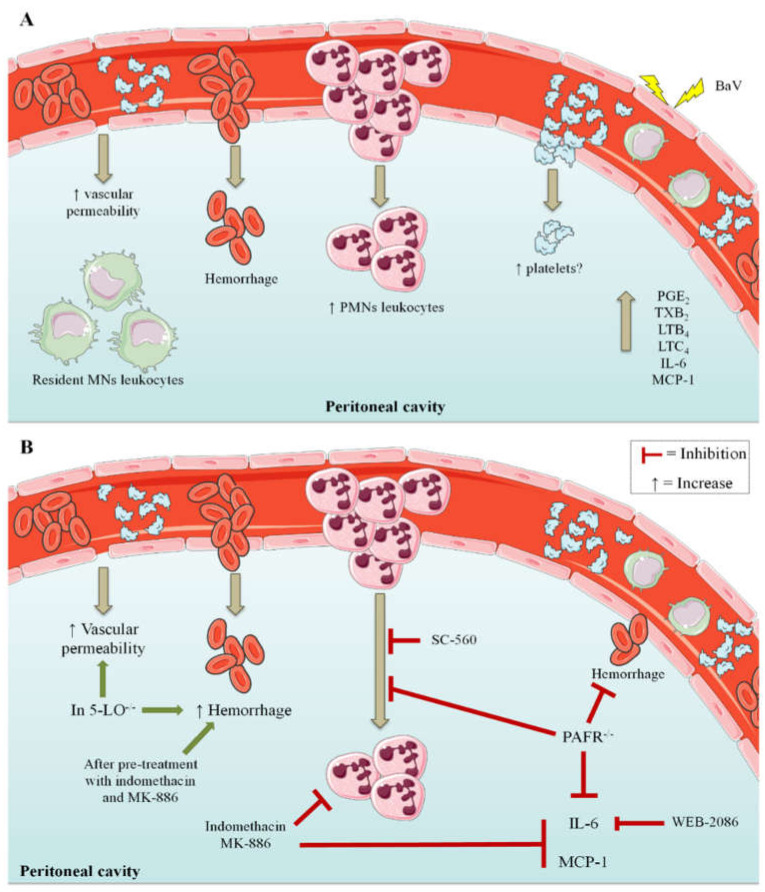
(**A**) BaV induced in vivo inflammation, characterized by increased vascular permeability and the number of PMN leukocytes in the peritoneal cavity, accompanied by the production of eicosanoids LTB_4_, LTC_4_, TXB_2_ and PGE_2_, as well as by the local and systemic production of IL-6 and MCP-1. Due to hemorrhage, it is possible that plasma components, such as platelets, have overflowed into the cavity, which could contribute to the inflammatory process. (**B**) After pharmacological interventions and the use of mice genetically deficient in 5-LO (5-LO^−/−^) and PAFR (PAFR^−/−^), it was possible to conclude that the inflammatory events induced by BaV are partially dependent on lipid mediators. The results also showed increased hemorrhage after pre-treatment with indomethacin and MK-886 and in 5-LO^−/−^ mice, which also had elevated vascular permeability. PLT: platelets; PMNs: polymorphonuclear; MNs: mononuclear. The schematic art pieces used in this figure were provided by Servier Medical art. Servier Medical Art by Servier is licensed under a Creative Commons Attribution 4.0 Unported License.

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
