# Peer review of "Bitis arietans Snake Venom Induces an Inflammatory Response Which Is Partially Dependent on Lipid Mediators"

_toxins, 2020, doi:10.3390/toxins12090594_

Round 1
Reviewer 1 Report
The manuscript entitled “Bitis Arietans Snake Venom Induces an Inflammatory Response which is Partially Dependent on Lipid Mediators” describes the contribution of lipid-derived mediators in the inflammatory events induced by B. arietans venom. The authors performed several experiments to reach the conclusion of the findings. However, I have some major concerns related to this manuscript.
Major comments
- The manuscript needs careful editing and revision with respect to English grammar, sentence structure, and wording. Proofreading is strongly recommended to ensure that the text is clear and understandable. For example - lines 53-54: plasma molecules, please specify or give an example; line 57: loss of function - what kind of function?; line 430: in the system “StepOne Plus” – Is it the real time PCR system? etc. Authors should check for italicizing where needed (ex. line 434 – no need to italicize for “ACK lysis buffer”). Some corrections have been identified, but not all have been listed in the review. Please check carefully if revision occurs.
- Why C57BL/6 and PAFR-/- were not tested for the vascular permeability?
- Figure 3 should be located before Figure 2. The supplementary Figures 1 & 2 should be included in Figure 3.
- To determine the genetic differences in the inflammatory processes after induction by B. arietans venom, authors should compare the inflammatory events between 129 SvE and C57BL/6 and it should be presented in the manuscript.
- Authors tested the effects of inflammatory events in C57BL/6 mice using 4 different anti-inflammatory drugs as shown in Figures 2, 4, 6, 7. It would be better to provide an individual inflammatory event of C57BL/6 mice treated with BaV in comparison to C57BL/6 treated with all inhibitors in the same graph. For example, in Figure 2A, the authors should merge Bav + WEM-2086, BaV + indomethacin, and BaV + SC-560 in the same graph. In addition, the authors should include a group of mice treated with inhibitor alone to validate the findings.
- Please include the reference that inject Evans blue through the retro-orbital plexus. Was this the first time that use this method of delivery of Evans blue? Two references (Sirois et al., 1988; Moreira et al., 2012) that authors cited used i.v. route for Evans blue injections.
- For clarity, authors should indicate mouse strain in the captions of every figures.
- Authors investigated the participation of COX-1 pathway downstream products in inflammation induced by BaV and TXB2 is a main product of COX-1 pathway. Why the production of eicosanoids LTB4, LTC4, PGE2, and TXB2 in mice pre-treated with SC-560 were not mentioned in the results section and not presented in Figure 7?
- Lines 281-284: Authors discussed that genetically deficient mice of the enzyme 5-lipoxygenase (5LO-/-) showed unexpected and different results when compared with C57BL/6 mice pre-treated with the 5-LO inhibitor, MK-886. The authors used 5LO-deficient mice, which are originated on a 129 SvE background as stated in the Materials and Methods section 5.4. How can the authors expect the same event occurs in a similar way in the genetically different lineage, C57BL/6 mice pre-treated with MK-886. Please clarify.
- The authors stated in the figure 8 caption, lines 347-349 that due to hemorrhage, it is possible that plasma components have overflowed into the cavity, mainly by the COX-1 pathway downstream products. This statement is not supported by the experimental data shown in Figure 7 and supplementary Figure 3. Please explain.
Minor comments
- The authors should go through all labels in the graphs. Figure 1A (y axis) is not clearly labelled. There are inconsistent fonts, symbols, and units in the manuscript ex. minute or min, hour or h, g/dl or g/dL, 5LO-/- or 5-LO-/-. Authors should use 30 min instead of ½ h or 0.5 h; 0.5 mg/kg instead of 0,5 mg/kg; NaCl instead of NaCL. Abbreviations should be defined in parentheses the first time they appear in the abstract, main text, and in figure captions and used consistently thereafter.
- Figure 7D: Authors should indicate what is the “C+” in figure caption?
Author Response
Dear Ms. Bonnie Yang, M.Sc
Managing Editor
Manuscript ID: toxins-894128
Title: Bitis arietans snake venom induces an inflammatory response which is
partially dependent on lipid mediators.
We wish to thank the Reviewers for the rather appropriate comments on our manuscript. The authors agree, and did correct and modify the text accordingly. The article was revised, and several adjustments were made in the English used, as indicated in the submitted manuscript.
The manuscript entitled “Bitis Arietans Snake Venom Induces an Inflammatory Response which is Partially Dependent on Lipid Mediators” describes the contribution of lipid-derived mediators in the inflammatory events induced by B. arietans venom. The authors performed several experiments to reach the conclusion of the findings. However, I have some major concerns related to this manuscript.
Major comments
1- The manuscript needs careful editing and revision with respect to English grammar, sentence structure, and wording. Proofreading is strongly recommended to ensure that the text is clear and understandable. For example - lines 53-54: plasma molecules, please specify or give an example; line 57: loss of function - what kind of function?; line 430: in the system “StepOne Plus” – Is it the real time PCR system? etc. Authors should check for italicizing where needed (ex. line 434 – no need to italicize for “ACK lysis buffer”). Some corrections have been identified, but not all have been listed in the review. Please check carefully if revision occurs.
Answer: We agree. The article was completely revised by a native of the English language, and several adjustments were made in the text, as indicated in the submitted manuscript. Unnecessary italicized terms were changed. We hope the text is now clear.
2 - Why C57BL/6 and PAFR-/- were not tested for the vascular permeability?
Answer: C57BL/6 and PAFR-/- lineages, with the same genetic background, developed early hemorrhage caused by BaV (0.5 mg/kg). Therefore, it was not possible to evaluate the increase in vascular permeability induced by the venom in these experimental models. However, using the 129 SvE and 5-LO-/- lineages, it was concluded that BaV can induce an increase in vascular permeability just after 10 min of venom inoculation. These information were included in the Discussion.
3 - Figure 3 should be located before Figure 2. The supplementary Figures 1 & 2 should be included in Figure 3.
Answer: To date, there are no studies showing the effect of BaV on the acute inflammatory process. In this sense, we initially chose to demonstrate that BaV is capable of inducing inflammation in vivo (Figure 1). Hence, the idea was to highlight the role of lipid mediators in this inflammatory process, using strategic inhibitors and genetically deficient mice. We demonstrated differences between the effects of BaV in 5-LO-/-mice from mice treated with the MK -886 inhibitor. Likewise, the results obtained with
PAFR-/- mice or treated with WEB-2086 were presented. We also choose to present the results obtained with the 129 SvE and 5-LO-/- mice, and the inhibitors, as complementary Figures 1 and 2, which were mentioned in the work. Thus, considering the reasoning mentioned above during the development of the work, we kindly ask for the understanding and permission of this Reviewer to maintain the current order of the figures.
4- To determine the genetic differences in the inflammatory processes after induction by B. arietans venom, authors should compare the inflammatory events between 129 SvE and C57BL/6 and it should be presented in the manuscript.
Answer: We agree with the observation. However, the aim of this study was to demonstrate the inflammation induced by BaV and the role of lipid mediators in this process. Thus, we ask for the understanding of this Reviewer why we did not carry out these experiments in this present work. Certainly, these pertinent observations will be considered in the studies which are now being performed on the inflammatory process, this time induced, exclusively, by the toxins isolated from BaV.
5 - Authors tested the effects of inflammatory events in C57BL/6 mice using 4 different anti-inflammatory drugs as shown in Figures 2, 4, 6, 7. It would be better to provide an individual inflammatory event of C57BL/6 mice treated with BaV in comparison to C57BL/6 treated with all inhibitors in the same graph. For example, in Figure 2A, the authors should merge Bav + WEM-2086, BaV + indomethacin, and BaV + SC-560 in the same graph. In addition, the authors should include a group of mice treated with inhibitor alone to validate the findings.
Answer: We appreciate and respect the good suggestion, however, the choice in the order of presentation of the results was justified in question 3. We understand that treatments with inhibitors can cause direct effects on animals, however, the focus of the work was to show the effect of these inhibitors on the inflammation induced by the venom. It is noteworthy that the animals in the control group were treated with the vehicle of inhibitors emulsified in NaCl, as mentioned in Material and Methods, to ensure that the observed effects are specifically due to the drugs used.
6 - Please include the reference that inject Evans blue through the retro-orbital plexus. Was this the first time that use this method of delivery of Evans blue? Two references (Sirois et al., 1988; Moreira et al., 2012) that authors cited used i.v. route for Evans blue injections.
Answer: We agree. The following reference was included in the Material and Methods' section: 54. Yardeni, T., Eckhaus, M., Morris, H. D., Huizing, M., & Hoogstraten-Miller, S., Retro-orbital injections in mice. Lab. animal 2011, 40 (5), 155-160.
7 - For clarity, authors should indicate mouse strain in the captions of every figures.
Answer: We agree. The mouse strain was included in all figure captions.
8 - Authors investigated the participation of COX-1 pathway downstream products in inflammation induced by BaV and TXB2 is a main product of COX-1 pathway. Why the production of eicosanoids LTB4, LTC4, PGE2, and TXB2 in mice pre-treated with SC-560 were not mentioned in the results section and not presented in Figure 7?
Answer: The project was in its final stage when it was considered pertinent to evaluate the role of COX-1 in BaV envenomation. In this way, the remaining experimental animals available were used to evaluate the effect of the COX-1 inhibitor (SC-560) on leukocyte migration and IL-6 and MCP-1 production in 4 hours. Thus, it was not possible to evaluate the production of LTB4, LTC4, PGE2 and TXB2 in 30 minutes, the selected period of stimuli for lipid mediators’ quantification in this work. However, as the inflammatory process related to B. arietans envenomation is a little explored topic, the results obtained with the COX-1 inhibitor, although in the absence of lipid mediators’ quantification, were included in the results.
9 - Lines 281-284: Authors discussed that genetically deficient mice of the enzyme 5-lipoxygenase (5LO-/-) showed unexpected and different results when compared with C57BL/6 mice pre-treated with the 5-LO inhibitor, MK-886. The authors used 5LO-deficient mice, which are originated on a 129 SvE background as stated in the Materials and Methods section 5.4. How can the authors expect the same event occurs in a similar way in the genetically different lineage, C57BL/6 mice pre-treated with MK-886. Please clarify.
Answer: We appreciate the question. To assess the participation of leukotrienes in BaV-induced inflammation, the experiments were conducted in control groups and in groups of animals whose leukotriene production was affected, either by genetic alterations or by pharmacological interventions. As low levels of leukotrienes were evaluated in both cases, it was expected that the results obtained would be similar, even in animals with different genetic backgrounds. This information was reinforced in the Discussion.
10 - The authors stated in the figure 8 caption, lines 347-349 that due to hemorrhage, it is possible that plasma components have overflowed into the cavity, mainly by the COX-1 pathway downstream products. This statement is not supported by the experimental data shown in Figure 7 and supplementary Figure 3. Please explain.
Answer: We agree. In fact, the sentence does not clearly convey the desired information. This statement is not based on results showed in the present work, but on literature data. To this, references 29, 33 and 35 were used, that emphasize the important role of platelets and TxA2 - the main metabolite of the COX-1 pathway - in the inflammatory response. For clarity, the sentence was rewritten.
Minor comments
11 - The authors should go through all labels in the graphs. Figure 1A (y axis) is not clearly labelled. There are inconsistent fonts, symbols, and units in the manuscript ex. minute or min, hour or h, g/dl or g/dL, 5LO-/- or 5-LO-/-. Authors should use 30 min instead of ½ h or 0.5 h; 0.5 mg/kg instead of 0,5 mg/kg; NaCl instead of NaCL. Abbreviations should be defined in parentheses the first time they appear in the abstract, main text, and in figure captions and used consistently thereafter.
Answer: We agree. The text and figures have been carefully revised and corrections have been made.
12 - Figure 7D: Authors should indicate what is the “C+” in figure caption?
Answer: We agree. This information was added in Figure 7D caption.
Reviewer 2 Report
The manuscript describes inflammatory responses to Bitis arietans envenomation. The authors also examined types of leucocytes accumulated and triggered production of eicosanoids, cytokines and chemokines. The result is very interesting and adds new valuable information to the field. However, there are several minor issues in the manuscript to be address as described below for the acceptance.
- The specific purpose of the study needs to be clearly described in Abstract. The rough description of experiments is also necessary in Abstract.
- Procedures to collect “local” and “systemic” samples needs to be described clearly in Methods.
- The third bars in Figs. 2, 4, 6 and 7 are inconsistently patterned between and within Figures and misleading. The patterns should be unified.
4 In Figure 7, the middle chart is missing the description of significance.
- Which is the evidence of the statement that “BaV stimulates PAFR coding genes” in L313 in Discussion?
- Fig. 8B, expressions of 5LO-/-, Indomethacin and MK-886 in the left part is misleading. Upper arrow meaning increase should not be used for genetically engineered condition and for exogenous molecules such as indomethacin and MK-886. Instead, they can be shown with red “inhibition” chart as shown in the right part of the figure. For example, the authors can show inhibition of hemorrhage by indomethacin and by MK-886. Inhibition of vascular permeability and hemorrhage by 5LO can be shown in the same way.
- (Minor typo) L415 and 416, superscript of “0” (zero) are used in “ËšC”. They need to be corrected as in L389.
- (Minor typo) L269, “c]” needs to be corrected as “c)”.
- (Minor typo) L493, “NaCL” needs to be corrected as “NaCl”.
Author Response
Dear Ms. Bonnie Yang, M.Sc
Managing Editor
Manuscript ID: toxins-894128
Title: Bitis arietans snake venom induces an inflammatory response which is
partially dependent on lipid mediators.
We wish to thank the Reviewers for the rather appropriate comments on our manuscript. The authors agree, and did correct and modify the text accordingly. The article was revised, and several adjustments were made in the English used, as indicated in the submitted manuscript.
The manuscript describes inflammatory responses to Bitis arietans envenomation. The authors also examined types of leucocytes accumulated and triggered production of eicosanoids, cytokines and chemokines. The result is very interesting and adds new valuable information to the field. However, there are several minor issues in the manuscript to be address as described below for the acceptance.
We appreciate the positive comments about our article.
1 - The specific purpose of the study needs to be clearly described in Abstract. The rough description of experiments is also necessary in Abstract.
Answer: We agree. The aim of the study and the description of experiments were included in the Abstract.
2 - Procedures to collect “local” and “systemic” samples needs to be described clearly in Methods.
Answer: We agree. These information were included in Materials and Methods.
3 - The third bars in Figs. 2, 4, 6 and 7 are inconsistently patterned between and within Figures and misleading. The patterns should be unified.
Answer: We agree. The third bars in Figs. 2, 4, 6 and 7 were standardized according to the type of inhibition used, which were divided in three groups: leukotrines inhibition, PAFR inhibition and prostanoids inhibition, which were dully cited in the respective figure captions.
4- In Figure 7, the middle chart is missing the description of significance.
Answer: Thank you for the observation. In fact, there was no statistical difference between the populations of mononuclear cells (MNs) of the experimental groups showed in figure 7. Despite the tendency to reduction the number of MN cells in the group inoculated with BaV, this number varied considerably between animals of the same group, making this difference not significant.
5 - Which is the evidence of the statement that “BaV stimulates PAFR coding genes” in L313 in Discussion?
Answer: We agree. The sentence was not formulated correctly and can lead to a wrong interpretation of the results. The authors wanted to refer to the participation of PAFR in the inflammation induced by BaV, since the animals genetically deficient in PAFR (PAFR-/-) showed a reduction in the inflammatory parameters evaluated in this study. PAFR participates in the inflammatory response induced by BaV, but it is not possible to say how the absence of PAFR could interfere in this process. Thus, this sentence has been rewritten.
6 - Fig. 8B, expressions of 5LO-/-, Indomethacin and MK-886 in the left part is misleading. Upper arrow meaning increase should not be used for genetically engineered condition and for exogenous molecules such as indomethacin and MK-886. Instead, they can be shown with red “inhibition” chart as shown in the right part of the figure. For example, the authors can show inhibition of hemorrhage by indomethacin and by MK-886. Inhibition of vascular permeability and hemorrhage by 5LO can be shown in the same way.
Answer: We agree. The Figure 8 and its caption were corrected to be clear as to show that there was an increase of hemorrhage after treatment with indomethacin, and MK-886 in 5-LO-/- mice, which also showed higher vascular permeability.
7 - (Minor typo) L415 and 416, superscript of “0” (zero) are used in “ËšC”. They need to be corrected as in L389.
Answer: We agree. The text was carefully reviewed and corrected.
8 - (Minor typo) L269, “c]” needs to be corrected as “c)”.
Answer: We agree. The text was corrected.
9 - (Minor typo) L493, “NaCL” needs to be corrected as “NaCl”.
Answer: We agree. The text was corrected.
Round 2
Reviewer 1 Report
Inconsistencies are still observed throughout the revised manuscript ex. minute or min; 0,5 mg/kg should be changed to 0.5 mg/kg (lines 175, 200, 238, 261, 282).
Author Response
Dear Ms. Bonnie Yang, M.Sc
Managing Editor
Manuscript ID: toxins-894128
Title: Bitis arietans snake venom induces an inflammatory response which is
partially dependent on lipid mediators.
1 - Inconsistencies are still observed throughout the revised manuscript ex. minute or min; 0,5 mg/kg should be changed to 0.5 mg/kg (lines 175, 200, 238, 261, 282).
The authors are grateful for the comments made in our article. The text has been completely corrected, and the terms have been standardized (minutes instead of min; 0.5 mg/mL instead of 0,5 mg/mL).
Minor adjustments to the text were also made, as described below:
Line 12: "the" has been added
Line 49: B. arietans envenomation (word order)
Line 83: B. jararaca and B. jararacussu venoms (word order)
Line 98: The word "studies" has been replaced by "researches”
Line 190: The word "important" has been deleted
Line 248: "Reduces" has been replaced by "reduced"
Line 275: Cox-1 expression (word order)
Line 328: "Leads to consideration" has been replaced by "may indicate"
We apologize, but the Figure 8 has been updated based on the comments made by Reviewer 2. Therefore, a new Figure 8 is being sent.